# A Bio-Social Model during the First 1000 Days Optimizes Healthcare for Children with Developmental Disabilities

**DOI:** 10.3390/biomedicines10123290

**Published:** 2022-12-19

**Authors:** Mark S. Scher

**Affiliations:** 1Pediatrics and Neurology, Rainbow Babies and Children’s Hospital, Case Western Reserve University School of Medicine, Cleveland, OH 44106, USA; mark.s.scher@gmail.com; 2Department of Pediatrics, Division of Pediatric Neurology Fetal/Neonatal Neurology Program, University Hospitals Cleveland Medical Center, Cleveland, OH 44106, USA

**Keywords:** first thousand days, bio-social model, healthcare disparities, gene-environment interactions, toxic stressors, developmental disabilities, maternal-placental fetal triad, neonate, educational neuroscience

## Abstract

Most children with developmental disabilities (DD) live in resource-limited countries (LMIC) or high-income country medical deserts (HICMD). A social contract between healthcare providers and families advocates for accurate diagnoses and effective interventions to treat diseases and toxic stressors. This bio-social model emphasizes reproductive health of women with trimester-specific maternal and pediatric healthcare interactions. Lifelong neuronal connectivity is more likely established across 80% of brain circuitries during the first 1000 days. Maladaptive gene-environment (G x E) interactions begin before conception later presenting as maternal-placental-fetal (MPF) triad, neonatal, or childhood neurologic disorders. Synergy between obstetrical and pediatric healthcare providers can reduce neurologic morbidities. Partnerships between healthcare providers and families should begin during the first 1000 days to address diseases more effectively to moderate maternal and childhood adverse effects. This bio-social model lowers the incidence and lessens the severity of sequalae such as DD. Access to genetic-metabolomic, neurophysiologic and neuroimaging evaluations enhances clinical decision-making for more effective interventions before full expression of neurologic dysfunction. Diagnostic accuracy facilitates developmental interventions for effective preschool planning. A description of a mother-child pair in a HIC emphasizes the time-sensitive importance for early interventions that influenced brain health throughout childhood. Partnership by her parents with healthcare providers and educators provided effective healthcare and lessened adverse effects. Effective educational interventions were later offered through her high school graduation. Healthcare disparities in LMIC and HICMD require that this bio-social model of care begin before the first 1000 days to effectively treat the most vulnerable women and children. Prioritizing family planning followed by prenatal, neonatal and child healthcare improves wellness and brain health. Familiarity with educational neuroscience for teachers applies neurologic diagnoses for effective individual educational plans. Integrating diversity and inclusion into medical and educational services cross socioeconomic, ethnic, racial, and cultural barriers with life-course benefits. Families require knowledge to recognize risks for their children and motivation to sustain relationships with providers and educators for optimal outcomes. The WHO sustainable development goals promote brain health before conception through the first 1000 days. Improved education, employment, and social engagement for all persons will have intergenerational and transgenerational benefits for communities and nations.

## 1. A Dual Diagnostic Approach for Accurate Diagnoses

Accurate clinical decisions require serial assessments across developmental time beginning before conception [1]. A dual diagnostic approach helps minimize cognitive biases [2] to achieve greater accuracy for more effective interventions. Horizontal analyses describe changing phenotypic form and function with maturation. Vertical analyses apply systems-biology within each developmental niche to identify etiopathogenesis from genetic through multi-systemic interactions [3] (Figure 1A,B). Immediate and delayed positive or negative G X E interactions will be discussed that apply principles of developmental neuroplasticity to distinguish brain health from disease. Career-long learning using this diagnostic approach benefits healthcare providers beyond formal training for more effective patient care, teaching, and research. New diagnostic advances will promote more effective neuroprotective interventions with improved outcomes.

Accurate clinical decision-making is particularly crucial for children who have increased risks for neurologic sequelae such as those with developmental disabilities (DD). DD consist of a constellation of functional deficits that are expressed beginning early in life, adversely affecting a child’s physical, learning, or behavioral performance [4]. Children exhibiting DD typically include sensory impairments of hearing and vision, cerebral palsy, seizures or epilepsy, attention deficit disorder, autism spectrum disorder, intellectual disability, and specific learning disorders. Different phenotypic combinations of DD present diagnostic and therapeutic challenges. Behavioral and mental health disorders are later expressed throughout childhood that further complicate healthcare delivery across the lifespan [5].

A maternal-child pair medical case history is discussed to highlight the importance of time-sensitive diagnostic assessments from pre-conception through adolescence for successful interventions. Maternal and childhood adverse effects were addressed. Revised educational plans through her school years integrated maternal-pediatric healthcare histories into successful educational experiences. A dual diagnostic approach is discussed that teaches recognition of the maturational potential of any child according to principles of developmental neuroplasticity applied to clinical decision-making and therapeutic interventions. All healthcare providers and educators require knowledge and skills to implement the same diagnostic and educational approaches for the most vulnerable children in LMIC and HICMD.

This child was previously included in a discussion of a diverse group of children who were diagnosed with genetic disorders [6]. Neurological diagnosis should consider G x E interactions with reassessments at older ages. Interplay of genetic disorders with acquired diseases and maternal-pediatric adverse effects are important considerations when applying a bio-social model of healthcare. Such a model will facilitate more successful outcomes when offered novel preventive, rescue, and reparative options for neuroprotection.

## 2. Maternal Autoimmune Disease and Prematurity

This maternal-child pair presented to healthcare providers and educators in a HIC. A serial diagnostic approach considered recognition of clinical phenotypes expressed by this woman, MPF triad, neonate, and child. Systems science application applied an understanding of etiopathogenesis pathways to the child’s neurologic sequelae. Time-dependent medical and scholastic resources were offered to preserve health while she completed high school education. Joint attention by her mother and healthcare providers addressed adverse childhood effects that potentially could reduce long-term benefits from interventions in response to early life expression of DD.

Mother was a 28-year-old gravida 2 para 1–2 woman with asymptomatic Sjogren’s Disease and class 3 obesity (i.e., body mass index > 40) prior to conception. She followed a weight reduction plan which included bariatric surgery before pregnancy and achieved a lower body mass index. Her healthcare providers noted the inherited risks for this pregnancy consisting of an older child with autism spectrum disorder and parental anxiety disorders. Mother received behavioral interventions to manage her anxiety both before and during her pregnancy. A high-risk maternal fetal medicine service managed her autoimmune disease. No immunosuppressive medications were recommended given she remained asymptomatic for Sjogren’s disease after conception.

An AV block on a fetal EKG was later documented during third trimester abdominal sonography at 28 weeks GA. Despite maternal treatment with digitalis, this dysrhythmia progressed to nonimmune hydrops fetalis at 32 weeks’ gestation secondary to cardiac failure. Labor induction and delivery was electively performed without complications after antenatal dexamethasone administration. No fetal distress occurred during delivery with a clinically stable female preterm newborn. She received 1- and 5-min Apgar scores of 6 and 7 without the need for advanced resuscitative interventions. Her anthropometric measurements were within appropriate gestational age ranges.

The child subsequently required neonatal intensive care for multiple system-specific complications. Multiple cardioversions for cardiopulmonary arrest and severe hypotension/bradycardia were required to treat repetitive sudden heart block during her first week of life until effective digitalis dosing was established. Pulmonary disease, anemia and nutritional supplementation required medical interventions during the neonatal time. Neonatal cranial sonography did not document intraventricular hemorrhage or periventricular leukomalacia.

Histological examination of the placenta at 10 days of life documented chronic villitis and maternal malperfusion lesions consisting of underdevelopment of the placental vasculature including decidual necrosis. These microscopic findings were supplemented by gross descriptions of a hyper coiled umbilical cord (2/cm) with a 1 cm marginal insertion.

The child initially expressed hypotonia which later resolved with expression of age-specific developmental reflexes. Bulbar dysfunction initially required gavage feedings because of poor oropharyngeal function. Gradual transition to oral feeding was achieved during her early months of infancy.

She required a permanent cardiac pacemaker at 11 months for persistent heart block for continued episodes of hypotension that remained unresponsive to medication adjustments. Cardiogenetic testing confirmed a long QT syndrome type II (G604S-KCNH-2), which was also confirmed for one parent.

Developmental delay with borderline microcephaly (5%) was documented on serial assessments during her first 2 years of life. Vision and hearing evaluations were normal. No seizures occurred. Early intervention therapies were provided to supplement and support parental care. Primary care included social work support to address adverse childhood effects. Developmental testing documented cognitive, motor, social and language deficits. Sufficient skill acquisition was achieved during her preschool yea to matriculate into mainstream classes. She advanced through primary and middle school grades assisted by individual educational planning.

She presented with status epilepticus associated with an acute hypertensive event at age 10 years. Brain lesions consistent with posterior reversible encephalopathy syndrome were noted on neuroimaging. Her pediatric intensive care successfully controlled seizures after stabilization of severe hypertension. With resolution of her encephalopathic state and renal dysfunction, she was discharged to home on antiepileptic medication. No neurologic regression occurred without further seizures. Antiepileptic medication was discontinued after a three-year interval.

She experienced cognitive and behavioral challenges into adolescence. Cooperation between family and the school’s multidisciplinary team of teachers and therapists offered revised interventions at school and home to address all medical challenges based on information provided to healthcare providers. Mental health care addressed her generalized anxiety and socialization challenges with educational accommodations and counselling. Functional neurological disorders included psychogenic seizures during adolescence were addressed with resolution. Dysautonomic symptoms of postural orthostatic tachycardia syndrome were recognized and managed with preventive interventions.

Individualized educational plans (IEP) accommodated this child’s expressive language delay, reading disability, hyperkinesis with short attention and lower limb spasticity, beginning during her preschool and primary school grades. Ambulation without ankle supports was achieved before her middle school years. She received psychotropic medications to manage anxiety without the use of stimulant medications. Serial neuropsychometric testing after age five years documented cognitive abilities within the normal range with subtest deficiencies in attention, reading and executive planning. She received educational accommodations through high school graduation. Behavioral interventions addressed her anxiety and socialization challenges which challenged her into adolescence. She successfully graduated high school with plans to pursue a nursing career.

## 3. Prenatal Contributions to Developmental Disorders

This next section reviews the systems science associated with the prenatal expression of diseases affecting this maternal-child pair. Applicability for effective horizontal and vertical approaches to diagnosis and treatment for a comparable woman in a LMIC and HICMD will be compared with the healthcare delivery received by this mother-child pair. Advocacy by healthcare providers regarding healthcare resources must consider diversity and inclusion to achieve optimal outcomes for healthcare and school readiness.

Any child with a genetic disorder exemplifies G x E interactions that can begin before conception with time-dependent expression that are influenced by adverse maternal and childhood acquired conditions. Phenotypic presentations may first appear during pregnancy or after birth [6]. For this maternal-child pair, identification of one parent with the same channelopathy, one sibling with autism spectrum disorder and parental anxiety emphasized the contributions of inherited vulnerabilities prior to conception.

Interactions between familial genetic risks and acquired conditions affected this maternal-placental fetal (MPF) triad and neonate with adverse consequences into early childhood. Etiopathogenetic pathways across developmental ages during her first 1000 days altered phenotypic expressions of her brain disorders. Global developmental delay with borderline microcephaly were early phenotypes. A hypertensive disorder was a later childhood expression of this genetic disorder. Cognitive/behavioral and mental health disorders became more dominant phenotypes during her adolescent development.

This child’s fetal presentation at 28 weeks GA with heart block occurred after activation of an inherited channelopathy by maternal autoantibodies despite mother’s asymptomatic Sjogren’s disease [7]. Autoantibodies were likely transferred to this fetus after approximately 12 weeks GA [7,8] with maturing placental function. Autoantibodies triggered this child’s genetic channelopathy through G x E interactions. Fetal cardiac failure occurred with further neonatal complications. She was subsequently diagnosed with a long QT syndrome Type II [9] that identified an etiology for the cardiac dysrhythmia. Multiorgan system dysfunction can be expressed by the long QT syndrome involving a ubiquitous HERG/Ky 11.1 channel defect [10]. This channelopathy affected two different systems during this child’s pediatric lifetime. Fetal cardiac disease presentation was followed by renal disease when she was 10 years old.

Without clinical symptoms and the risk for adverse effects of corticosteroid use, mother received no treatment after conception. She alternatively was followed by a high-risk obstetrical service that monitored her based on maternal levels of care relevant to the this HIC health system.

Pregnancy management for a woman in a LMIC or HICMD with asymptomatic Sjogren’s disease might advocate for more proactive medical interventions to maximize chances for favorable outcomes despite less available prenatal and neonatal services. Earlier steroid treatment despite no maternal clinical signs may be the more practical therapeutic option. Treatment prior to conception or early first trimester for any woman identified with autoimmune disease might avoid or lessen the severity of fetal cardiac disease, hydrops fetalis and prematurity. Long QT syndrome may still present at older postnatal ages. However, with greater brain maturity, less severe neurologic sequalae may result because of increased resilience to brain injury. Benefit from immunomodulators other than steroids [11] in the future may also offer greater protection from the adverse inflammatory effects from pre-clinical autoimmune disease with less unacceptable side effects.

Preventive healthcare training for maternal care by physicians, nurse practitioners and midwives in a LMIC and HICMD must prioritize preventive care to offset less access to expensive high-risk maternal fetal medicine and neonatal intensive care services. WHO guidelines for maternal and neonatal levels of care [12], have emphasized the importance of such anticipatory care. This proactive approach to diagnosis and treatment should be directed to the most prevalent diseases of women during their reproductive years such as genitourinary infections, hypertensive disorders, and diabetes beginning before conception. Aggressive interventions should include mental health care to address vulnerabilities based on adverse maternal and family adverse effects.

Other non-infectious inflammatory conditions accompanied this mother’s Sjogren’s disease. Her obesity and anxiety worsened similar inflammatory disease pathways subserved by this autoimmune disorder. Suboptimal placental implantation and development began 2–8 days after conception given this exaggerated pro-inflammatory state. The placenta’s shared genetic endowment between parents and fetus [13] more likely also expressed this channelopathy as abnormal early placental development. Suboptimal placental formation then negatively influenced neural plate appearance at 17–19 days post conception. Multiple fetal systems subsequently may be affected by ongoing placental dysfunction across trimesters. Cardiovascular and nervous systems were specifically affected for this child.

Use of corticosteroids, weight reduction and treatment for anxiety before conception in LMIC and HICMD could help delay or avoid trimester-specific placental diseases for a percentage of MPF triad This preventive approach could avoid fetal cardiac failure, hydrops fetalis and prematurity.

Prematurity is a complex adverse outcome. Trimester-specific G x E interactions to the MPF triad involve placental dysfunction resulting in prematurity [14]. Suboptimal placental implantation shortly after conception resulted from mother’s abnormally heightened pro-inflammatory state. Impaired trophoblastic precursors in the first trimester contributed to second and third trimester placental vascular maldevelopment. Defective remodeling of spiral arteries altered villous development with uteroplacental insufficiency contributing to a premature birth.

Current fetal surveillance tests cannot fully document placental disease pathways affecting many MPF triads. Training of healthcare providers particularly in LMIC and HICMD must anticipate these adverse effects of placental-cord pathology without the availability of fetal surveillance testing to document abnormalities. Chronic or acute diseases are often undetectable using fetal sonographic assessments of fetal heart patterns, Doppler flow indices and biophysical scores to assess fetal well-being. Appropriate measures of fetal growth and function does not exclude risks from placental disease with adverse outcomes to the fetal nervous system.

## 4. Integrating Placental-Cord Pathology into the Diagnosis of Developmental Disabilities

Training of placental-cord development and pathogenesis enhances the provider’s understanding of time-dependent etiopathogenesis that contributes to sequelae such as DD. Placental-cord disease pathways are associated with diseases expressed by the woman, MPF triad and neonate. This perspective improves clinical decision-making, effective interventions, and prognosis. Despite normal fetal surveillance testing, postnatal pathological examinations may document trimester-specific disease pathways. Future prenatal structural and functional placental imaging studies will offer more effective diagnosis and interventions to improve outcomes for affected MPF triads [15] before fetal or neonatal diseases are expressed. Clinical pathology assessments should be requested and reviewed by obstetrical and pediatric healthcare providers.

Three placental disease mechanisms cumulatively contribute to an understanding of adverse effects to the fetal brain. Knowledge of these disease entities are essential for maternal-pediatric healthcare providers to advocate for improved medical delivery particularly in LMIC and HICMD.

Maternal immune activation (MIA) was the initial pathologic process that presented shortly after conception for this MPF triad. MIA represents immune intolerance between the embryo and the mother [16]. Suboptimal development at the maternal-fetal uterine interface occurs when the blastocyst improperly attaches to the inner surface of the uterus with adverse effects on placental precursor cell populations. While fetal loss may not occur, defective placental function beginning at 2–8 days following conception contributes to adverse effects on multiple developing fetal systems throughout the pregnancy.

MIA is associated with reduced energy substrate delivery, waste removal, and deprivation of essential growth factors as embryonic-fetal maturation advances into second and third trimesters [17]. MIA contributes to multisystemic MPF triad diseases, including the developing fetal brain. Multiple precursor neuronal and glial cell populations within the embryonic-fetal brain are at risk for impaired proliferation, differentiation, and migration within transient brain structures such as the ganglionic eminence and subventricular zone [18]. Adverse effects later develop in marginal and early subplate zones. These early abnormal stages in brain maturation precede the formation of a more mature cortical mantle. MIA has been associated with a spectrum of adverse neurologic outcomes including autism spectrum disorder, cerebral palsy, epilepsy, and behavioral/cognitive disorders as expressed by this child [19]. (Figure 2A,B). Chronic villitis was a pathologic biomarker for this pregnancy noted after birth that can be associated with first trimester MIA. Villitis has been interpreted as the result of a fetal allogenic graft rejection [20] given mother’s immune intolerance of her implanted embryo.

Ischemic placental syndrome (IPS) was the second placental disease that developed later during this pregnancy given postnatal gross and histological placental-cord findings. Prenatal sonographic measurements described preserved fetal growth for this child without fetal growth restriction. However, abnormal placental angiogenesis from IPS can still reduce placental blood flow to the fetus without current fetal surveillance detection. Defective spiral artery remodeling after 12 weeks GA results from abnormal precursor trophoblastic cellular development that begins during early first trimester [14,21] with resultant IPS (Figure 3). Malperfusion lesions of the placental vasculature were later documented on placental histopathology that are associated with IPS. These lesions are associated with chronic reduced placental flow to the fetus during second and third trimesters which contributed to fetal cardiac failure requiring elective delivery and premature delivery.

Marginal cord insertion [22] with hyper coiling [23] represented first trimester anomalous umbilical cord development that further reduced blood flow delivery to the fetus in addition to the placental malperfusion lesions. Narrowed vascular diameters with high vascular resistance from a marginally inserted hyper coiled cord were markers of reduced blood flow that limited oxygen, glucose, and growth factor delivery. Fetal cardiac failure progressing to hydrops fetalis from long QT syndrome also reduced fetal cerebral blood flow over the month before preterm delivery despite digitalis treatment. Neurologic sequelae are more likely associated with immune and non-immune forms of hydrops fetalis [24] with G x E interactions. This child’s inherited channelopathy contributed to her presentation of non-immune hydrops fetalis [25] after trimester-specific placental malperfusion resulting in cardiac failure.

Fetal inflammatory response (FIR) is a third pathologic mechanism that potentially affected placental structure and function in addition to MIA and IPS (Figure 4). Often undetectable with asymptomatic women, acute and chronic forms of FIR are later identified by postnatal histopathological examination applying international classification criteria [26]. FIR can be associated with infectious and/or noninfectious etiologies. Placental villitis was a postnatal marker for the proposed FIR Type II [27] that may have impaired this MPF triad during the first trimester. The disease pathway associated with FIR Type II resembles MIA as a threatened rejection of a semi-allogeneic embryo-fetus given toxic stressor pathways [28].

Maternal autoantibodies (i.e., untreated Sjogren’s disease), the metabolic syndrome (i.e., pre-conception maternal obesity and a potential pre-diabetic state) and elevated endogenous steroid and norepinephrine production (i.e., toxic stressors from ongoing maternal anxiety) cumulatively contributed to this mother’s non-infectious pro-inflammatory state into the third trimester.

FIR type I is the more predominant etiopathogenetic mechanism closer to delivery. Two disease pathways have been identified with FIR Type I, contributing to fetal brain injuries. Asphyxia occurs from inflammatory mediator-induced vasoconstriction within the placenta/cord and fetal brain blood–brain barrier vasculature resulting from direct effects by pathogens and abnormal cytokines on blood vessel caliber size and contractile responsivity. A second cytokine-associated mechanism with FIR Type I contributes to this asphyxial neuronal damage. Mitochondrial dysfunction within the cytosol and altered genetic expression within nuclei result in injury from cytokines with cell death or dysfunction in surviving neurons [29].

Fetal brain maldevelopment and/or injury therefore potentially result from the cumulative effects of MIA, IPS and FIR to a vulnerable MPF triad across three trimesters. Altered embryonic-fetal brain structures during the first trimester impair multiple neuronal cell precursors within transient region with abnormal proliferation, differentiation, and migration. Abnormal neuronal connectivities later alter the cortical mantle during the second and third trimesters (Figure 5) with abnormal structure and function. These three pathologic placental mechanisms also increase susceptibility for postnatal brain injuries.

Balanced development of excitatory and inhibitory neuronal precursor activities is required for normal fetal brain maturation and function. An altered excitatory/inhibitory ratio resulting from these placenta-cord diseases contribute to altered brain circuitries expressed as DD, mental health disorders and epilepsy [30].

Present standards of maternal levels of care [31] for this maternal-child pair in a HIC still lack sufficient sensitivity and specificity on fetal surveillance testing to detect MIA, IPS or FIR during many pregnancies. These limitations will be particularly challenging for providers in LMIC and HICMD. Training regarding the risks from placenta-cord pathology during pregnancy help anticipate the importance of postnatal pathological assessments.

## 5. Neonatal Contributions to Developmental Disorders

Four “great neonatal neurological syndromes” comprise most children who require neonatal neurocritical care [1]. Neonatal encephalopathy, seizures, and stroke present for both preterm and full-term neonates Encephalopathy of prematurity (EP) is expressed by vulnerable preterm infants and comprise the highest percentage of a neonatal survivors requiring intensive care.

G x E vulnerabilities of this maternal-child pair exemplified how prenatal and postnatal diseases contributed to EP and later DD.

Developmental and destructive brain processes define EP with trimester-specific disease pathways that later influenced postnatal disease factors [32]. Risks for neurologic sequalae such as DD with educational challenges were associated with EP from multi-systemic conditions of prematurity. Fetal and neonatal consequences to brain development resulted from the effects of EP [33].

Over 80% of nearly 15 million preterm children are born in LMIC [34]. Increased risks for EP consequently exist. Global public health programs such as The Every Newborn Action Plan by the WHO stress the need to prevent prematurity and stillbirth by improving maternal and neonatal care [35]. Aggressive antenatal dexamethasone if preterm birth is anticipated is a cost saving example applicable in LMIC and HICMD hospitals to lower mortality and reduce the cost per disability-adjusted life-years [36].

These efforts will lower the prevalence and severity of EP for vulnerable MPF triads such as those exposed to healthcare disparities in LMIC and HICMD.

Postnatal complications of prematurity further increased risks for EP following fetal cardiac failure with hydrops fetalis. An elective premature delivery with later postnatal medical complications occurred. Multiple pointes de torsade required emergent cardioversions during her first week of life to treat her cardiac decompensation. These events represented cumulative postnatal risks for injury from cerebral hypoperfusion. Additional adverse conditions of prematurity included pulmonary disease, anemia and gastrointestinal immaturity which contributed to reduced oxygen and nutrient delivery.

Current conventional postnatal cranial sonography and conventional brain MRI studies [37] documented intraventricular hemorrhage and periventricular leukomalacia as gross markers of major destructive lesions. Present neuroimaging tools lack precision to detect more pervasive brain lesions [38]. Future multimodal testing using quantitative and functional imaging will more precisely document fetal brain and placental development [39,40,41,42] as prenatal markers for EP. Novel structural and functional neuroimaging biomarkers will more accurately detect a wider spectrum of cerebral dysmaturity and injury associated with EP. These innovative prenatal imaging procedures will be applied to future pharmacologic protocols designed to treat and preserve healthy placental and fetal brain function [38], and reduce the effects of EP and childhood sequalae.

Such diagnostic and therapeutic advances will be first available to MPF triads and neonates in HIC where there is greater access to research protocols. Research protocols in LMIC and HICMD require strategies for best practices to deliver technologies to those who are most vulnerable with healthcare disparities. Maternal-pediatric hospitals in these countries and regions require research efforts that compare different preventive, clinical and wellness study protocols to best deliver healthcare to women and children [43] to reduce the incidence of EP.

## 6. Childhood Contributions to Developmental Disorders

Developmental delay during this child’s first 1000 days represented postnatal clinical expression of previously acquired fetal and neonatal brain disorders. Early interventions supported by parental commitment offered greater healthcare and educational opportunities. Preschool expression of DD was replaced during school years with neurocognitive and mental health challenges as she matured into adolescence.

Primary pediatric care partnering with early intervention programs need to offer all children opportunities to address medical and developmental needs. Support for early diagnosis and rehabilitative interventions must begin during the first 1000 days. Challenges are greater with more limited access to preventive, clinical and wellness resources in LMIC and HICMD. Community-based primary care with the ability to provide early intervention programs must be accompanied by wellness programs to minimize adverse childhood effects.

Communicable and non-communicable diseases during childhood further complicate healthcare with adverse effects educational achievement. An acute hypertensive crisis with a cerebrovascular complication for this child at 10 years of age presented a new medical challenge. Her neurologic presentation of posterior reversible encephalopathy syndrome [44] was associated with her inherited genetic disorder. The long QT syndrome was associated with her acute renal disease presentation. Specific organ system disorders from long QT syndrome may present at different ages, triggered when new diseases or adverse conditions present [45].

Sudden systemic hypertension potentially alters cerebral blood flow preferentially to posterior brain regions where cerebral vasculature is highly susceptible to the adverse effects of increased systemic blood pressure. Persons with hypertensive disorders throughout their lifespan can experience brain injuries from this disease. The etiopathogenesis responsible for the clinical presentation of encephalopathy and seizures impairs function within the neurovascular unit (i.e., blood–brain barrier). Risks for stroke and/or hemorrhage are known complications. A percentage of affected persons, however, exhibit transient or reversible neuroimaging findings as the acute clinical symptoms resolve. This child fortunately did not experience permanent injuries expressed by more severe sequelae.

Intensive care management for this child included emergent treatment for status epilepticus and acute hypertensive crisis. Seizures were successfully controlled with resolution of neuroimaging findings on follow-up brain MRI images. Medication discontinuation without seizure reoccurrence after a three-year course of antiepileptic drug treatment was achieved. No apparent regression of her neurologic abilities resulted from these medical events.

Healthcare systems in LMIC and HICMD need to prioritize continuity of medical delivery when childhood diseases present that require hospitalizations. Management of childhood medical complications present greater challenges when there is less access to advanced hospital-based medical care. WHO guidelines offer suggestions for improved hospital services for children confronted with acute illnesses in LMIC and HICMD [46]. Healthcare systems specific to a region or nation confront the healthcare workforce with challenges. Community-based, business, and governmental efforts need to offer funding for hospital-based care for the most vulnerable children.

## 7. Adolescent Brain Maturation with Mental Health Disorders

Brain maturation during adolescence involves accelerated synaptogenesis that reflects adaptative or maladaptive neuroplasticity. More complex brain connectivities are needed for optimal adult neurologic function. Increased neural network complexity is required to achieve sophisticated cognitive reasoning and social intelligence [47] required for successful workplace performance and positive social interactions.

Mental health challenges accompany DD which may intensify during adolescence. This child’s anxiety, mood instability, and problems with social interactions were adolescent expressions of abnormally increased synaptogenesis which potentially could further limit her adult performance. She experienced functional neurologic disorders and mental health challenges at home and in school despite lessening of earlier life DD. Continuity of psychiatric interventions and social supports by her family helped her through graduation. Continued mental health care by family and healthcare providers will be needed as she prepares for a career, establishes social relationships, and confronts life challenges [48].

Mental illness expressed during childhood more likely is associated with brain disorders during the first 1000 days. Children in a LMIC or HICMD may not experience the same degree of recovery as the child presented for this discussion. Neuron-specific cell processes may result in atypical brain development starting during fetal life with more severe expressions of childhood neurocognitive/behavioral disorders including mental health disorders into adulthood. Public health resources must recognize the importance of mental health care and social problems to address childhood adverse effects. These risks are greatest for those with DD who experience healthcare disparities [49].

Two longitudinal/cross-sectional cohorts present results applicable to the importance of preclinical diagnosis of mental health disorders. Comparisons of neuroimaging and genetic data sets suggested more accurate prediction of mental health disorders prior to clinical expression. Region-specific growth of the cerebral cortex was documented using detailed neuroimaging protocols based on large pediatric cohorts. These studies preceded mental illness and general psychopathology as expressed throughout childhood [50]. These MRI signatures were correlated with genome-wide association data sets. This longitudinal cohort correlated neuron-specific genetic markers to quantitative MRI profiles. Similar methodology has been reported in Danish longitudinal imaging-genetic cohort comprised of individuals who have familial risk for schizophrenia and bipolar disease [51]. Both studies suggest that effective preventive treatment strategies can be achieved before mental health disorders are expressed.

These neurodiagnostic advances will benefit children with DD who may also present with mental disorders. Availability for such diagnostic testing is needed in LMIC or HICMD for children at highest risk for mental health disorders accompanying DD.

Healthcare policy therefore must advocate for mental healthcare in these vulnerable populations. Unfortunately, cultural misperceptions of DD and mental illness impede these efforts. Criminalization and religious demonization of people with these disorders result in exclusion of children with DD from receiving adequate healthcare and education. Support for families must respect religious, ethnic, and racial perspectives to properly direct interventions. Appropriate services need to be prioritized by community-religious leaders, policy makers, healthcare providers, and educators.

## 8. Continuum of Maternal-Pediatric Risks for Neurologic Sequelae

The maternal-child pair used for this discussion stresses the continuum of prenatal and postnatal risks that began with maternal autoimmune disease, obesity, and anxiety. Fetal cardiac disease and complications of prematurity were adverse outcomes associated with these conditions. MPF triad phenotypes more broadly reflect multiple maternal diseases (e.g., hypertension [52], diabetes [53], and obesity [54]), placental diseases (e.g., maternal immune activation [16], ischemic placental syndrome [55], and fetal inflammatory response [29]), and fetal diseases (e.g., prematurity [56] and intrauterine growth restriction [57]) with risks for neurologic sequelae beginning before conception. Knowledge by healthcare providers in LMIC and HICMD are needed to deliver effective medical interventions to those who are most vulnerable.

Etiopathogenetic mechanisms from these diverse diseases and conditions affect the developing MPF triad with harmful effects on the developing fetal nervous system. Oxidative stress, inflammatory response and hypoxia-ischemia are disease pathways that synergistically alter neuronal cell populations and connectivities [58]. Diverse expressions of neurologic sequelae may result including preterm survivors [59] with DD.

Neonatal encephalopathy, seizures, and stroke are expressed by only a minority of preterm and full-term survivors [1]. Most children with these “great neonatal neurological syndromes” experience antepartum and peripartum diseases from diseases affecting the MPF triad before labor and delivery. Intrapartum sentinel events contribute to brain injury for only a small percentage of survivors.

Postnatal system-specific diseases also contribute to sequalae in vulnerable populations. Congenital heart disease [60], persistent pulmonary hypertension of the newborn [61] and sepsis [62] are prevalent illnesses that contribute to neonatal morbidities.

Medical management of the great neonatal neurological syndromes present a greater challenge in LMIC and HICMD where more limited access to transport and neonatal intensive care delivery exists. These healthcare systems require healthcare provider training and resources to better identify MPF triad risk and advocate for effective neonatal intensive care to lower complications.

The “silent” majority of healthy newborns will present during childhood despite uneventful pregnancies and neonatal presentations. Children with neurologic sequelae such as DD may be undetected based on prenatal or neonatal conventional testing that cannot identify the full range of brain disorders with later childhood clinical expression [1]. Communicable and noncommunicable illnesses experienced throughout childhood from infections, accidents and multisystemic diseases either initiate or worsen neurologic disorders. G x E interactions further influence the diversity and severity of clinical expression [6] of neurologic disorders throughout the lifespan [63], Dementias, stroke, epilepsy, and neurodegenerative diseases during later adulthood are also associated with the developmental origins of brain disease occurring before the first 1000 days [63].

This “silent” majority in an LMIC and HICMD present unique challenges within these healthcare systems. Continuity of medical care require necessary resource delivery in communities to respond when a child experiences communicable or noncommunicable illnesses in the context of adverse childhood effects. Support for primary-care family and pediatric practices must recognize when neurologic sequelae more likely may occur. Training of healthcare providers in rural and urban centers include nurses, therapists, and physicians to anticipate and address risks. Budgetary support to train, employ and equip primary care professionals must be a high priority. Regionalized higher levels of medical care within hospital systems must offer emergency transport when advanced medical care is required to hospital facilities equipped for neonatal and pediatric intensive care.

## 9. Educational Neuroscience Applicable to Children with DD

Professionals in the school setting require familiarity with the principles of educational neuroscience when designing individual educational plans to address children who express DD [64]. An interdisciplinary approach by the school team includes teachers, therapists, and school psychologists [65]. Serial neuropsychometric assessments using standardized testing tools supplemented by qualitative observational information of the child in the school setting constitute the best methodologies to design and adjust educational plans through high school [64] (Figure 6). Applying neurodiagnostic information such neuroimaging studies [66] of children with learning deficits can assist in developing effective educational plans. Three examples describe the variable onset, type, and severity of DD across developmental time when designing appropriate educational strategies for those most vulnerable to DD particularly in LMIC or HICMD.

Children with a preschool presentation of autism spectrum disorder often present with language delay in the context of state dysregulation, lack of social responsivity and sensory sensitivities during the first 1000 days. Women may have experienced preconception and periconception diseases that reduced immune tolerance of blastocysts on placental implantation with development of MIA. Anomalous neuronal precursor cell populations later result that contribute to an imbalance of excitatory and inhibitory brain circuitries as brain structures mature over the first 1000 days [67].

Children who were subjected to the negative effects of MIA on placental function during the first half of pregnancy [68,69,70] will later more likely express behavioral characteristics of autism spectrum disorder. This medical history can assist in preschool scholastic planning to design appropriate educational plans to anticipate deficits in language development, sensory processing, cognition, and social communication expressed by children with autism spectrum disorder as they enter primary school grades.

The preterm survivor who survives IPS during the second half of pregnancy is the second example of how specific DD may require a focused approach to an individual educational plan. Children born premature more likely experience learning disabilities expressed as deficits in visual processing and mathematical skills in addition to problems with reading comprehension and spelling [71,72]. Brain localization of these deficits include vulnerable posterior brain regions injured by IPS that subserve these cognitive functions. These scholastic challenges are often clinical expressions of prenatal and postnatal effects from EP. Educators need to apply neurodiagnostic evaluations when addressing these cognitive deficits. This information will help design more appropriate educational plans to address specific learning disabilities.

Peripartum and neonatal complications experienced by the near-term and full-term neonate is the third example applicable to educational neuroscience. More mature fetal and neonatal brain regions express more region-specific brain dysfunction when confronted with peripartum and neonatal illnesses expressed as encephalopathy, seizures, or stroke. These children later express a broad range of DD affecting motor, language, cognitive and other social-adaptive functions [73]. Lesions in more mature neuronal connectivities within frontal lobe and limbic structures of the term neonate later express as deficits in executive function, attention, and mood stability in the context of cerebral palsy and associated motor deficits. Teachers should consider a child’s medical history when designing appropriate lesson plans that address these specific scholastic challenges.

Educational neuroscience training is paramount for teachers in LMIC and HICMD to provide children with DD high-quality scholastic experiences. Knowledge of childhood neurological disorders requires familiarity of the healthcare providers diagnostic evaluations when designing effective educational intervention plans. Educators must advocate for funding support to maintain these programs through all grades. Recognition of ethnic, religious, gender, sexual orientation differences must be part of these programs to optimally incorporate all children into the educational system

## 10. Healthcare Delivery for Children with DD

Healthcare and educational opportunities for children with DD are curtailed when socioeconomic, racial, ethnic, and cultural diversities are not addressed in a timely manner. Cognitive biases by providers impede effective clinical decision-making for children with these challenges. Substandard healthcare delivery and greater risks for sequelae [74] disproportionately affect children with DD. Inequitable healthcare delivery and inattention to toxic stressors present social problems with health consequences for children with DD within their communities [75].

Most children with DD consequently live in LMIC and HICMD [76] where health and educational services are not as readily available. These children are at greatest risk for suboptimal healthcare delivery, educational failure, and a poorer quality of life than children without DD. Application of epidemiological datasets specific to regional, national, or global perspectives by health policy makers remain underreported, particularly during the first 1000-day perspective. Addressing healthcare disparities for women and children will facilitate medical care equity and improve statistical estimates to better allocate needed resources.

Worldwide estimates suggest that 94% of children with DD live in LMIC [76]. Nearly a quarter to a third of a billion children and adolescents worldwide experience DD where healthcare and educational resources are limited [77]. Even in a HIC such as the United States where 6% of the world’s population of children with DD reside, a disproportionate majority live where less healthcare access and greater toxic stressors prevail for children and families in medical deserts within a HIC (HICMD). Over a third of the US population lives in a county where there is less than adequate access to pharmacies, primary care providers, hospitals, trauma centers, and/or low-cost health centers [78]. These women, children and families lack healthcare access and are more vulnerable to toxic stressors beginning before conception. Data from the National Health Interview Survey reported increasing prevalence of DD among US children aged 3–17 years between 2009–2017 [79], following similar worldwide trends as reported using functional or statistical models in LMIC [80]. Increasing numbers of children with DD emphasize the need for a bio-social model of care that advocates for healthcare beginning before the first 1000 days for the most vulnerable women and children in both LMIC and HICMD.

Time-dependent expression of maternal-placental-fetal-neonatal phenotypes later present as different neurologic disorders across the lifespan beginning with DD. Etiopathogenetic mechanisms affecting the developing brain during the first 1000 days more likely result in permanent maladaptive neuroplasticity before full clinical expression of neurologic sequelae. Permanent dysfunction more likely results after exposure to adverse toxic effects to the woman before conception with cumulative consequences to the MPF triad, neonate, and child less than two years. Effective programs to address adverse childhood effects are extremely important [81] that exacerbate the effects of DD. However, such family-based support must be preceded by care for women during their reproductive years as well as during and in between pregnancies.

Principles of educational neuroscience integrates neuroplasticity concepts into preschool developmental interventions for children who are recognized as the most vulnerable for DD. Teachers [64] during school years incorporate these preschool templates into individual educational plans for the most challenging children with revisions as challenges are addressed. Both developmental care providers and educators require familiarity with the developmental plasticity concept of critical-sensitive time periods. Altered neuronal connectivities during the first 1000 days later remain as permanent cognitive/behavioral deficits across the lifespan. More effective educational experiences can only be offered once these children are identified by healthcare providers with accurate diagnoses. Educational neuroscience must be part of teacher training to help reduce the global burden of neurologic disorders across the lifespan [63] by improved educational experiences to prepare for adulthood. This will enhance employment opportunities and social relationships in their communities.

## 11. A Social Contract for Healthcare Providers and Educators

The WHO recently revised world health statistics in the context of the disastrous effects from the coronavirus (COVID) pandemic [82]. Seventeen sustainable development goals (SDGs) were proposed in 2015 that advocate for optimal health and wellbeing. for persons in all nations. As with the COVID pandemic, responses to future threats to world health from diseases, conflicts, and environmental conditions will continue to challenge achievement of these goals. Implementation of these SDGs include the integration of medical and educational strategies for all children. Effective partnerships between families and healthcare providers begin with a social contract before the first 1000 days that then contribute to educational success through cooperation between families and teachers. Global healthcare advocacy combined with effective educational strategies will benefit all children, including those with DD in LMIC and HICMD. These children can then experience greater adult potential [83,84].

Lawmakers and health policy leaders in both HIC and LMIC must adopt bio-social models of care that advocate for optimal care for all persons across the lifespan. This social contract by healthcare providers [74,85] benefits the woman, child, family, and community before, during and between pregnancies. This bio-social model of healthcare prepares children for effective health and educational experiences and an improved quality of life.

The mother-child pair chosen for this review represented a family living in a comparatively resource-rich community. Her parents’ access to the internet and community services provided information and guidance to receive maternal and pediatric healthcare, minimize adverse childhood effects and plan educational interventions as their daughter’s medical and school challenges were identified. Teachers and therapists effectively applied knowledge of educational neuroscience applicable to her specific scholastic needs. Healthcare and educational services remained available for this family throughout high school.

Healthcare strategies must address cultural, financial, and political barriers to care [86] beginning during the first 1000 days. This requires a strong social contract between provider, educator, and patient with continuity over time [6,74,75]. Resource-limited regions within LMIC and HICMD offer substantially less access to medical and school services for girls, women, children, and families before, during and between pregnancies. Medical problems enhanced by endogenous/exogenous toxic stressors consequently have a greater negative impact on maternal, MPF triad, neonatal and childhood health outcomes. Disease expression with G x E interactions across developmental time will consequently result in higher mortality and morbidity. Greater rates of pregnancy-related maternal and infants’ death disproportionately affect people of color and lower socioeconomic status [87,88]. Greater numbers with fetal demise or neonatal stillborn status as well as survivors with more severe neurologic sequelae result. These challenges later adversely influence the child’s education throughout school years. Greater risks for neurologic and mental health disorders expressed by children are more likely to negatively impact communities into adulthood. Maintaining a social contract by healthcare providers and educators with families will optimize school readiness and improve life-course health and quality of life for all children.

Principles of ontogenetic adaptation help explain long term effects of positive and negative neuroplasticity [89]. Time-dependent effects on neurodevelopment of these children during the first 1000 days will more likely have permanent effects across the lifespan and for successive generations. Maladaptive neuroplasticity more likely result for those children with DD in LMIC and HICMD. Intergenerational and transgenerational effects [90] must be anticipated and addressed.

With advances in medical care into adulthood, longer life expectancy for older survivors with DD requires a greater priority for inclusion regarding optimal educational attainment during childhood to achieve better employment and societal interactions as adults [77]. There is a moral imperative by providers to honor a social contract to best serve the most vulnerable persons. Support of the sustainable development goals of the WHO specifically SDG 4 apply to children and adults with DD, most of whom live in LMIC and HICMD.

As estimates for life expectancy increase for people with DD, acute and convalescent medical services and quality home care delivery must be anticipated to assist aging families and their proxies. Adults with moderate intellectual disability and cerebral palsy may potentially live into their fifties, assuming stable health without accidents or chronic illnesses. Milder clinical expressions of disability now experience similar life expectancy estimates as healthy populations without DD [91,92,93].

Communicable and noncommunicable diseases disproportionately affect more women and children in LMIC and HICMD. Maternal health consequences after genitourinary infections, hypertensive disorders, and diabetes mellitus are examples of prevalent maternal conditions that adversely affect fetal brain health presenting as diseases of the MPF triad and neonate. The great neonatal neurologic syndromes of encephalopathy, seizures, stroke, and EP are postnatal expressions of trimester-specific diseases after birth. Childhood neurologic disorders such as DD are further influenced by ongoing diseases or adverse conditions.

## 12. Equitable Global Healthcare Policy for Women and Children

Diversity and inclusion for healthcare and education delivery are essential for optimal outcomes. Integrating developmental origins and life course theories into clinical and public health practices [63] will serve children into adulthood. Effective maternal-pediatric healthcare delivery will sustain brain health across the lifespan by effectively treating diseases and toxic stressors. This investment will reduce medical expenses, increase productivity through employability, and improve a nation’s wealth and wellness.

The WHO sustainable development goals advocate for all women and children. Transactional models to achieve these goals require healthcare policies that integrate the continuity of maternal and pediatric healthcare beginning before conception when first addressing a girl’s reproductive health. These goals are later integrated with advocacy for pregnancy planning and optimal maternal preventive care during and between pregnancies. Communities require resources provided by public health programs for effective nutrition, health care, housing, and sanitation to maximize life-course health during and between each pregnancy.

Preventive healthcare education and resources require support by a nation’s public health infrastructure that can effectively provide medical and educational services to children and families that recognize and overcome geographic-cultural barriers. With public health efforts, effective educational plans can be initiated at the earliest ages to optimize the school performance of a child with DD, applying educational neuroscience by teachers recognizing ethnic, racial, and cultural context. These actions will benefit the most vulnerable with healthcare disparities in LMIC and HICMD.

While these efforts potentially have significant global impact, wide gaps remain between preclinical diagnosis of DD and implementation of interventions for women of reproductive age between and during their pregnancies. The WHO sustainable development goals recently highlighted brain health across the lifespan to promote adaptive neuroplasticity starting during critical-sensitive periods. This emphasis on brain health must advocate for women, children, and adolescent health, with interventions that begin during the first 1000 days and continue into adulthood [83].

Every Newborn Action Plan presented by the WHO further emphasizes strategies to reduce neonatal morbidity and mortality. Success of this plan also advocates for prenatal health of women and MPF triads to maximize the benefits for neonates who require resuscitation, transport, and neonatal intensive care [12].

International cooperation among government and nongovernmental organizations must advocate for a woman’s right for optimal health care and educational services for her children within their community. Such advocacy must be maintained from childhood through adulthood for healthy children and those with DD. Future research protocols need to address maternal and pediatric health disparities by applying public health initiatives that educate women and families to advocate for resources to sustain health into adulthood. Effective partnerships will improve the woman’s health and quality of life throughout life beyond her reproductive years as well as for her children and successive generations.

Future biomarkers can potentially apply genetic-metabolomic, neurophysiologic and neuroimaging technologies [94] for pre-clinical diagnosis of children with DD. These diagnostic advances will provide opportunities to assess the efficacy of preventive, rescue and reparative neurotherapeutics as illnesses are experienced across childhood [1]. These scientific advances must be accessible for all persons. Accurate medical diagnoses, treatments and educational interventions require that geographic, political, and cultural barriers must be overcome, particularly in LMIC and HICMD.

## Figures and Tables

**Figure 1 biomedicines-10-03290-f001:**
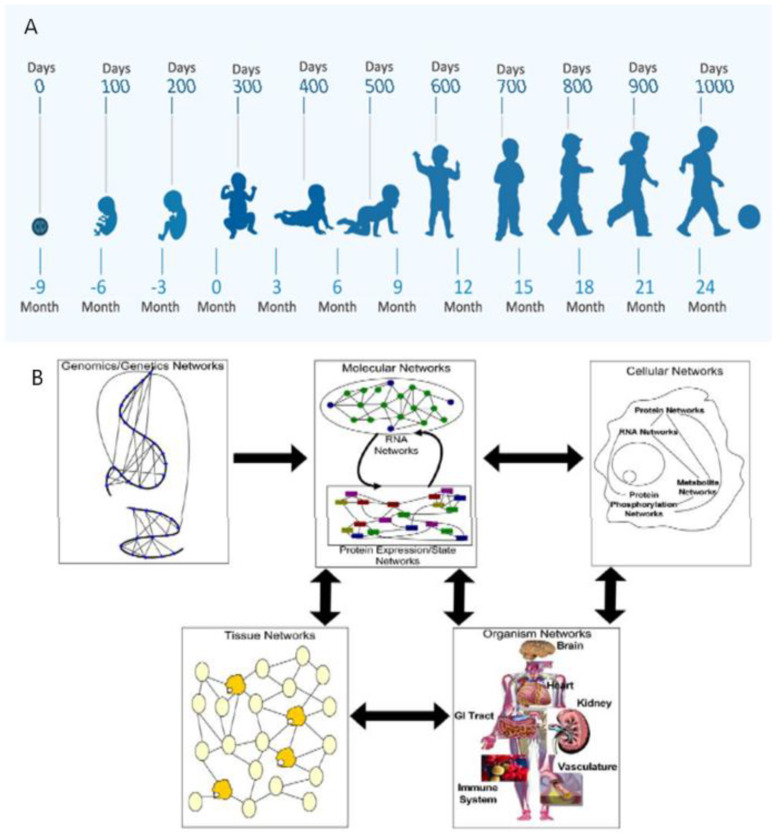
(**A**) The horizontal approach to the diagnostic process across maturing phenotypes over the first 1000 days. The WHO program diagram depicts the first 1000 days from conception through two years of age that illustrates changing clinical expressions. (**B**) The vertical approach to the diagnostic process depicted as the flow of information across biological systems through a hierarchy of networks. Each panel highlights a different set of networks at play in a biological system. Genomics networks represent interactions among DNA sequences that may give rise to longer-range as well as more local chromosomal structures that modulate gene activity, in addition to inducing synergistic effects on higher-order phenotypes. Genomics networks drive molecular networks composed of RNA, protein, metabolites, and other molecules in the system. Molecular networks are components of cellular networks in which the complex web of interactions among these networks gives rise to the complex phenotypes that define living systems. Tissue networks comprise cellular networks that are clearly influenced by the molecular and genomics networks, and organism networks comprise tissue networks that are clearly defined by the component cellular and molecular networks. Complex phenotypes like disease emerge from this complex web of interacting networks, given genetic and perturbations to the system [3].

**Figure 2 biomedicines-10-03290-f002:**
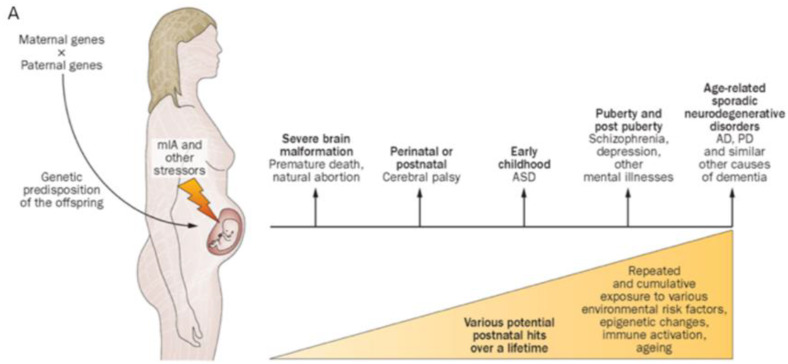
(**A**) Proposed causal chain of events with maternal immune activation in humans, leading to a wide spectrum of neuronal dysfunctions and behavioral phenotypes observable in the juvenile, adult, or aged progeny. Abbreviations: AD, Alzheimer disease; ASD, autism spectrum disorder; MIA, maternal immune activation; PD, Parkinson disease [19]. (**B**) Pathways for neurodevelopmental resilience and susceptibility to maternal immune activation are depicted, induced by infectious or noninfectious stimuli with variable effects on the offspring. Whereas a substantial portion of offspring are resilient to maternal immune activation and do not acquire overt pathologies, neurodevelopmental sequelae occur in susceptible offspring. The neurodevelopmental consequences among the latter are heterogeneous and span a range of neurological and psychiatric disorders with varying temporal onsets (as indicated by the corresponding lengths of the arrows and the broken-lined circles). Note that the illustrated pathways among susceptible offspring represent risks but not deterministic relationships [16].

**Figure 3 biomedicines-10-03290-f003:**
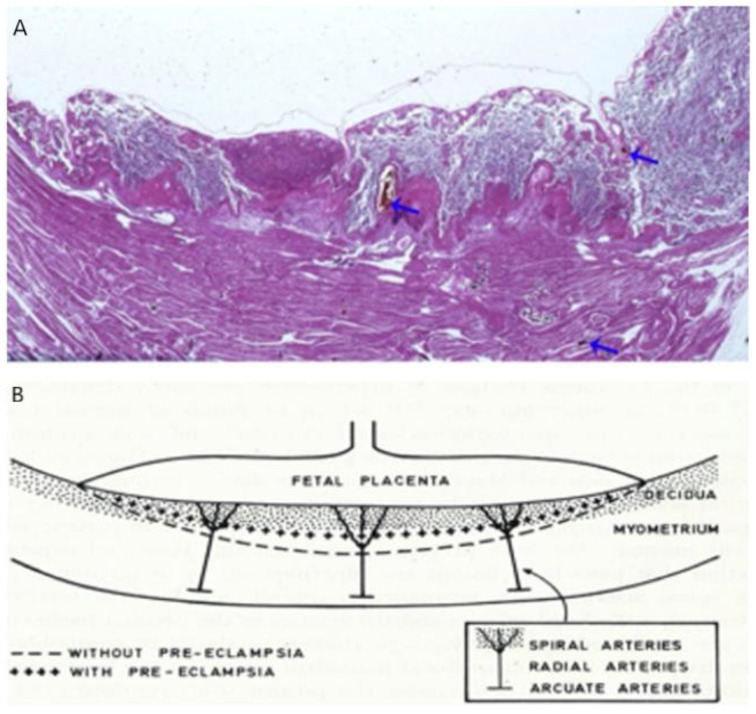
(**A**) Microscopic appearance on H& E staining of an infarcted zone of a uterus with placenta in situ in severe preeclampsia. The intravascular injection of Chinese ink via the uterine artery appears as patent radial and spiral arteries in the central part of the placental bed (blue arrows on the right and center) but absent arteries are associated with a large intervillous thrombus (blue arrow on the left). (**B**) Schematic diagram of placental vessels’ transformation showing normal (bottom line) and defective deep placentation. [21].

**Figure 4 biomedicines-10-03290-f004:**
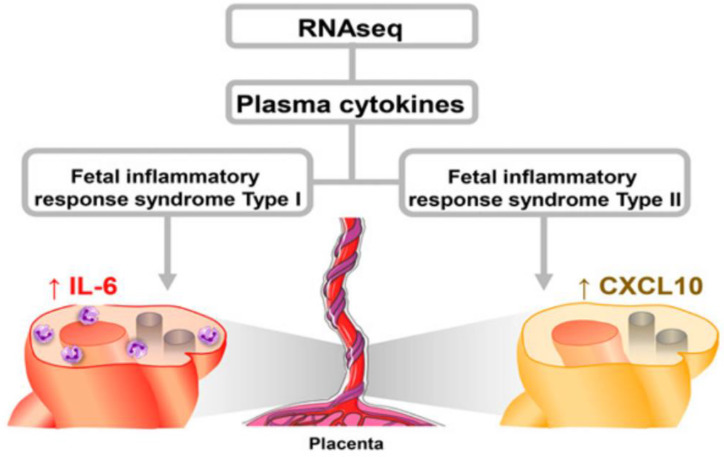
Two types of fetal inflammatory response have been described based on statistical analyses comparing the results of RNA sequencing from human cord blood samples of preterm neonates diagnosed with fetal inflammatory response syndromes type I and type II after birth associated with placental-cord histopathological findings. Placental examinations were performed by pathologists blinded to the clinical histories. FIRS type I was characterized by an upregulation of host immune responses, including neutrophil and monocyte functions, together with a proinflammatory cytokine storm and a downregulation of T cell processes. IL 6 was the representative cytokine that was excessively released. This process was correlated with acute and chronic lesions of chorioamnionitis and funisitis. In contrast, FIRS type II comprised a mild chronic inflammatory response involving perturbation of HLA transcripts, suggestive of fetal semi-allograft rejection. CXCL 10 was the representative chemokine that was excessively released. This process was depicted by findings of chronic chorioamnionitis, villitis or deciduitis. Abbreviations: IL 6, interleukin 6; CXCL 10, chemokine 10 [27].

**Figure 5 biomedicines-10-03290-f005:**
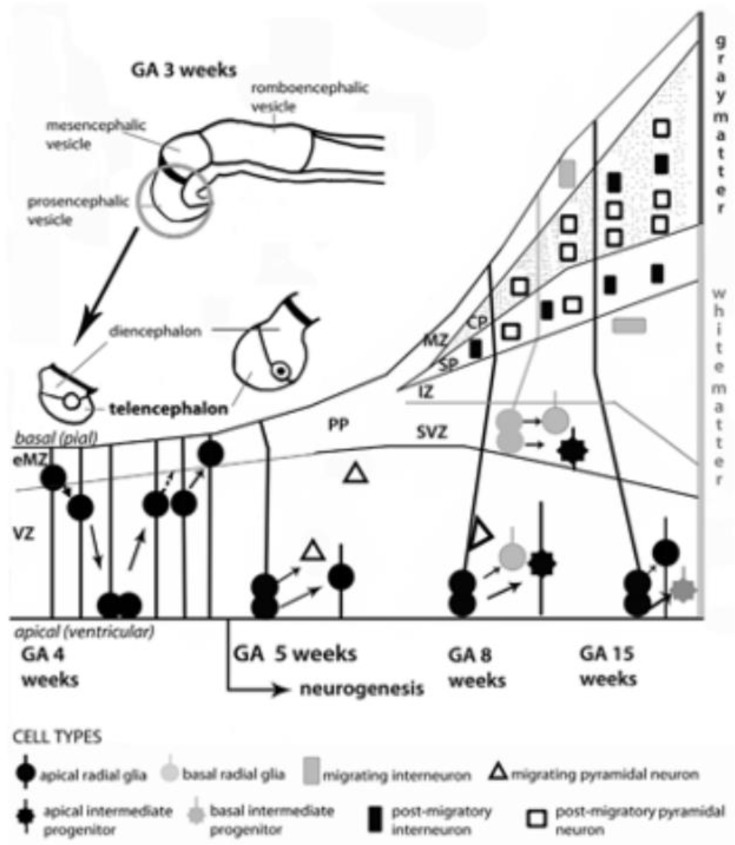
The neuro-ontogenic process starts at gestational age (GA) weeks 2–3 with the constitution of the neural tube. At GA week 4, the rostral portion of the neural tube forms the prosencephalic, mesencephalic and romboencephalic vesicles. The prosencephalic vesicle then forms two vesicles that are destined to become the telencephalon and the diencephalon (thalamus, hypothalamus, and other structures). The schematic diagram represents the development of telencephalon. Initially, the telencephalic primordium is constituted by dividing neuroepithelial cells (often called neural stem cells), characterized by interkinetic nuclear migration, which form the ventricular zone (VZ). There, one type of cells, the radial glial cells, are particularly prominent and distinctive. At GA week 4, they undergo early exponential proliferation increasing the number of progenitor/intermediate progenitor cells and the thickness of VZ. Early born neurons are interneurons which move within the marginal zone (MZ) and intermediate zone (IZ). The MZ will eventually form cortical layer I and, for some authors, MZ could be identified also before the cortical plate (CP) formation as early MZ (eMZ). The IZ is a cell-sparse compartment and exists before the appearance of the CP. Radial glial cells and intermediate progenitor cells can be classified in two distinct subpopulations: apical, resident in VZ with bipolar fibers, and basal, which delaminate from VZ with unipolar basal fiber. Apical progenitor cells in the VZ and basal progenitor cells in subventricular zone (SVZ) are considered the main source of pyramidal neurons. Around GA week 5, the neurogenesis begins, and the neuronal precursors proliferate rapidly within the VZ. The neurons located in the first recognizable cortical layer, known as the preplate (PP), form the earliest synaptic connections. PP is a transient structure present before the appearance of the CP. Around week 7, the PP cells contribute to the subplate (SP) which remains below the CP after its formation and contains post-migratory pyramidal neurons and interneurons. Moreover, the accumulation of basal progenitor cells creates a distinct new compartment above the VZ, the SVZ. Here, they divide and are considered an additional source of intermediate progenitor cells. By GA week 8, radially migrating neurons from VZ and SVZ initiate the development of the layered CP forming from inside to outside. For example, pyramidal neurons eventually migrate outward along the radial glial cells [18].

**Figure 6 biomedicines-10-03290-f006:**
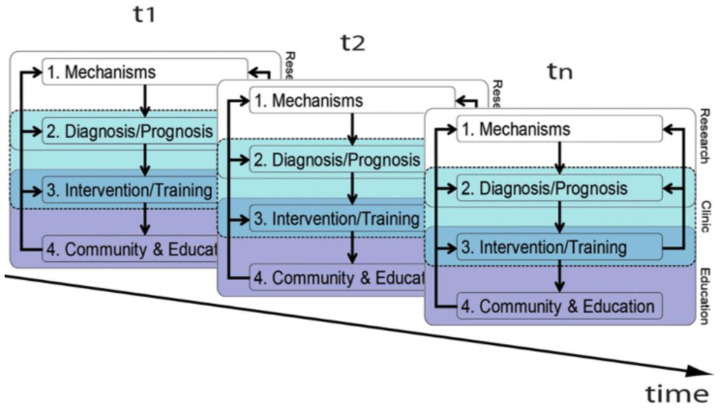
Diagram depicts the theoretical translational framework for the use of educational neuroscience by the educator to develop effective individual educational plans. Results of brain structural and functional testing are offered by clinicians to educators regarding diagnosis and prognosis of the maturing child through the school years. Interpretation of these findings are based on knowledge of developmental neuroscience research applied to clinical decision making. Interventional strategies are developed by the interdisciplinary educational team of teachers, psychologists and therapists based on this information at multiple time points. Lessons plans are revised as a function of advancing school grades. These teaching strategies are generally applied across a community’s school system calibrated to resource-availability [66].

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
