# Peer review of "A Bio-Social Model during the First 1000 Days Optimizes Healthcare for Children with Developmental Disabilities"

_biomedicines, 2022, doi:10.3390/biomedicines10123290_

Round 1

Reviewer 1 Report

Thank you for the opportunity to review this manuscript. The author have presented a review of bio social model for optimization of health care for children with developmental disabilities juxtaposed a clinical case. The overall review is cogent but the back and forth transition from case to review across the paper appears to interfere with the course of the review. The case narrative does not flow unseemly with changing heading of the review. I would recommend summarizing the case prior to the review with its separate heading. This may allow for an unencumbered flow of the review part of the paper and make it a better read overall. 

Author Response

see my comments 

Reviewer 2 Report

This is a good review of the need for early interventions and consideration of extending early intervention to pregnancy and pre pregnancy

There is a focus on inflammatory contributions to the genetic and other environmental factors - with a significant emphasis on immunological aspects.

There is some mention of the social aspects but not quite as balanced.

Overall raising the complexity of developmental considerations when working with children and developmental disorders. Not new but summarised well and raises the additional complexities of the neonatal and pregnancy inflammatory additions for consideration

Author Response

see my comments
